# Utilization of government healthcare services by adult leprosy patients in the Western Province, Sri Lanka

**Nadeeja Roshini Liyanage**[1]*, **Mahendra Arnold**[2], **Supun Wijesinghe**[3]

**1** National Programme for Tuberculosis Control and Chest Disease, Ministry of Health, Colombo, Sri Lanka, **2** Quarantine Unit, Ministry of Health, Colombo, Sri Lanka, **3** Health Promotion Bureau, Ministry of Health, Colombo, Sri Lanka

* nadeejaro@gmail.com

## Abstract

### Background

The leprosy services utilization by the patients at the clinic and field level should be high to achieve the target of eliminating leprosy as a public health problem in Sri Lanka. Furthermore, assessing patient and health system delay of a diagnosis and patient knowledge on disease are of equal importance to reveal the accurate picture.

### Methods and findings

A descriptive cross-sectional study was conducted to assess the utilization of government healthcare services by 672 adult leprosy patients in Western Province (WP). Paucibacillary patients diagnosed at least six months and above, and Multibacillary patients diagnosed at least 12 months and above were selected by consecutive sampling method. An interviewer-administered questionnaire (IAQ) was used for data collection.

Clinic utilization by leprosy patients was 87.8%. The mean patient-related delay (time taken from the onset of symptoms to the encounter of a doctor/health facility for the first time) was 16.8 months and health care system delay (time taken from the date of clinic registration to start of treatment) was 21.2 days. The overall delay was 17.5 months. Services provided by the Medical Officer of Health (MOH) office for families affected with leprosy was known by 53.8% (n = 298) of patients. Majority of family contacts were examined at the hospitals (n = 299, 44%), 30.8% (n = 207) by the Public Health Inspectors (PHI) and 7% (n = 46) at the MOH offices. PHIs had visited 56.7% (n = 401) of the patient's houses and 54% (n = 363) had received health education by PHI. Mean knowledge score was 50.7 (SD = 17.9). More than half (57.9%, n = 389) of the study sample had a good or very good knowledge level.

### Conclusions

Utilization of clinic services was satisfactory. However, a considerable patient-related delay was found. Half of the patients were aware of available field services and a majority of contact screening was conducted at hospitals. Patient knowledge on leprosy was satisfactory.

**Data Availability Statement:** Data cannot be shared publicly due to the patient confidential information included in it. The datasets used and/or analysed are available from the Ethics Review

Committee of the Medical Research Institute, Colombo 08. (Project No 55/2017. Email: erc_mri2016@hotmail.com.

**Funding:** The author(s) received no specific funding for this work.

**Competing interests:** The authors have declared that no competing interests exist.

## Author summary

Leprosy is a chronic progressive bacterial infection caused by *Mycobacterium Leprae*. It mainly attacks the skin, peripheral nerves in the hands, feet and eyes, causing numbness or weakness of the affected area and resulting in chronic morbidities such as vision impairment, limb disability, trophic ulcers and nerve involvement. Leprosy is assumed to be spread via the respiratory system through nasal droplets. It is commonly found among people living in poor socioeconomic conditions. Therefore to prevent getting complications, early diagnosis and treatment is mandatory. Our aim was to identify any delays in diagnosis and treatment; whether patients are utilizing the existing facilities and to assess patient knowledge of the disease and treatment. Knowledge of the healthcare workers to identify the disease and patient knowledge to continue the treatment is of equal importance to reduce the case load, reduce the diagnostic delay and prevent developing complications. To find the reasons why the services are not utilized and communicate the true picture to the decision makers are the objectives of this study.

## Introduction

Health care utilization is the measure of the population's use of the health care services available to them [1,2]. Health care utilization and health status are indicators to measure how efficiently a health care system provides services in a population. Service utilization is the extent to which people are making use of the services that already exist in the community or at an organization [3,4,5]. The importance of assessing utilization of a health service is that the policymakers will be aware of the existing services and underutilized services which will help them to decide which services should be expanded or which should probably be discontinued. It will also assist health care providers in planning future programs and using their resources more effectively. Assessing service utilization is beneficial since it can be monitored and followed for any changes over time [6,7]. The knowledge of the patient affects their health seeking behavior [8]. If the patient has less knowledge of symptoms of the disease, mode of transmission and treatment, it will lead to delayed presentation to the health system or might end up with deformities.

Leprosy is a neglected tropical disease and approximately 2000 cases are reported annually in Sri-Lanka [9]. Assessing the healthcare service utilization of leprosy patients and finding the reasons behind underutilization will enable to explore the true status of the health care delivery and facilities. Following the integration of leprosy services to general health services in 2001/2002, leprosy cases are managed at dermatology clinics conducted in base hospitals and above with a Consultant Dermatologist [9]. Central Leprosy clinic (CLC) functions under Anti-Leprosy Campaign and is situated in the premises of the National Hospital of Sri Lanka (NHSL). It is specially designed to treat only leprosy patients from all over the country. It provides comprehensive patient care including diagnosis, management, skin smear testing, physiotherapy services, counselling services and wound care. Apart from CLC, there are three other dermatology clinics functioning in the NHSL. There is a leprosy clinic in the Prison Hospital Welikada, which provides services to the imprisoned patients. Furthermore, the Lady Ridgway Hospital for Children provides services for children. Additionally, leprosy treatment is provided in the dermatology clinics of Base hospitals and above throughout the country.

With regards to field-based services, leprosy became a notifiable disease in 2013 and contact tracing was started in 2014 in Sri Lanka. Medical Officer of Health (MOH) is responsible for

preventive health services at the community level. The preventive services include reduction of active transmission of disease, lowering delayed presentation, improving quality clinic services, providing rehabilitating services, staff training and monitoring of the programme.

Since there are few studies carried out in the past in the proposed area, findings of the present study will provide the latest data in Sri Lanka on the utilization of services and knowledge of the disease of adult leprosy patients. The findings will be beneficial for healthcare planners in deciding future strategies, policy-making and implementation of new control activities.

## Methods

### Ethics statement

Ethical approval was obtained from the Ethics Review Committee of the Medical Research Institute, Colombo, Sri Lanka. Informed written consent was obtained from each participant prior to data collection. The consent of participants under the age of 18 was obtained from the participant as well as from the parent or guardian.

A descriptive cross-sectional study was carried out to assess the utilization of healthcare services by 705 adult leprosy patients who attended government leprosy clinics in the Western Province (WP) in the year 2018. A person 15 years and above was considered as an adult leprosy patient [10].

The service utilization in the clinic setting was operationally defined as "Attendance of leprosy patients to clinics regularly and continue treatment as prescribed". For example, if an MB patient from the point of diagnosis to one year period attended the clinic for a minimum of 12 visits and PB patient from the point of diagnosis to six month period attended the clinic of minimum six visits, that patient's clinic utilization was considered as 100%.

In 2015, the highest percentage (38%) of leprosy patients was reported from Western Province. Therefore WP was selected to conduct the study [9]. Among the nine provinces in Sri Lanka, WP reported the largest share of the population (28.7%) and it consists of Colombo, Gampaha and Kalutara districts [11]. The study was carried out in all leprosy clinics, dermatology clinics and households of leprosy patients. The study population consisted of all Paucibacillary adult leprosy patients diagnosed at least six months and above and, all Multibacillary adult leprosy patients diagnosed at least twelve months and above in the WP.

Clinic leprosy register was used as the sampling frame. Details of all the registered leprosy patients in the clinic were available in this register. Clinic leprosy register is maintained in every institution where leprosy patients are being treated. Consecutive sampling method was applied to select patients attending the clinic. Clinic attendance of patients was checked by refereeing to their clinic records.

Since the number of leprosy patients attending clinics is low, all eligible patients were recruited to achieve the required sample size. Adult leprosy patients not attending the clinic were traced from the details available in the clinic leprosy register and their data collection was carried out at the household level.

Interviewer administered questionnaire (IAQ) was used to assess the service utilization (S1 Text). It consists of socio-demographic data, information on disease and treatment, questions to assess the clinic and field utilization, and patient's knowledge of leprosy. When assessing the knowledge on the disease and its transmission, marks were allocated in a way that each correct response got one mark, and no marks for incorrect or don't know responses. The total score was calculated and the final marks were given as a percentage of the maximum marks allocated for the component. The knowledge score was categorized into four categories; less than 25% -very poor knowledge, 25%-49%—poor knowledge, 50–74%—good knowledge, and 75–100% -very good knowledge.

## Results

There were 705 patients, of which 33 patients declined to consent to participate due to time constraints. Hence, the response rate was 95.3%. The study was carried out among 672 patients who attended the government leprosy clinics in the WP. Of the sample the highest number of patients (n = 534, 79.5%) were resided in the Colombo district and out of the 119 patients (17.7%) were from the Colombo Municipal Council area (CMC). Majority of patients (n = 290, 43.2%) were attended to the CLC (n = 290, 43.2%).

A large majority of the study sample were in the age group 30–44 years (n = 221, 32.9%), males (n = 419, 62.4%) and married (n = 489, 72.8%). The majority of the study sample consisted of patients educated up to Ordinary Level Examination (n = 269, 40%), in paid employment (n = 307, 45.7%) and those who were having a monthly family income of Rs 20,000-Rs.40,000[108–216$] (n = 328,48.8%) (Table 1).

## Disease type

The study sample consists of a higher proportion of Multibacillary patients (n = 473, 70.4%). Large proportion 607 (90.3%) of patients had initially experienced pale or reddish patches over

**Table 1. Distribution of patients by selected demographic and socioeconomic characteristics (n = 672).**

| Demographic & socioeconomic characteristics | Frequency (n) | Percentage (%) |
|---|---|---|
| **Age Group(years)** | | |
| 15–29 | 145 | 21.6 |
| 30–44 | 221 | 32.9 |
| 45–59 | 175 | 26.0 |
| 60 or more | 131 | 19.5 |
| (Mean = 43.8, SD = 16.1, Median = 42, Range = 15–94) | | |
| **Sex** | | |
| Male | 419 | 62.4 |
| Female | 253 | 37.6 |
| **Marital status** | | |
| Married | 489 | 72.8 |
| Unmarried | 152 | 22.6 |
| Widowed | 23 | 3.4 |
| Divorced | 08 | 1.2 |
| **Level of Education** | | |
| No schooling | 50 | 7.4 |
| Up to Grade 5 | 90 | 13.4 |
| Up to Grade 8 | 104 | 15.5 |
| Up to Ordinary Level | 269 | 40.0 |
| Up to Advanced Level | 144 | 21.4 |
| Tertiary education | 15 | 2.2 |
| **Employment Status** | | |
| Unemployed | 276 | 41.1 |
| Self-employed | 89 | 13.2 |
| Paid employment | 307 | 45.7 |
| **Monthly Family Income** | | |
| Less than Rs 20000(108$) | 236 | 35.1 |
| Rs 20000–40000(108–216$) | 328 | 48.8 |
| Rs 40001-60000(217–326$) | 69 | 10.3 |
| More than Rs 60000(326$) | 39 | 5.8 |

 

the skin with loss of sensation as a symptom of leprosy. Over half of the patients obtained initial consultation at a government hospital (n = 393, 58.5%).

## Patient and clinic-related delay

The mean patient-related delay (time taken from the onset of symptoms to the encounter of a doctor/health facility for the first time) is 16.8 months with a median of 8 months (Table 2). Mean time duration from presentation to a doctor to diagnosis is 18.6 days. Mean time duration from diagnosis to starting treatment is 2.7 days. Mean health care system delay (time taken from the date of clinic registration to start of treatment) is 21.2 days. Mean overall delay (time taken from the onset of symptoms to the start of treatment) is 17.5 months with a median of 8 months. Mean duration of patient-related delay was high among MB (17.9 months, SD = 36.5) patients compared to PB patients (14.1 months, SD = 21.7). Mean health care system delay was high among PB patients (31.2 days, SD = 234.8) when compared with MB patients (17.1 days, SD = 139.2). The overall delay was high among MB patients (18.5 months.SD = 36.6) when compared with PB patients (15.2 months, SD = 23.1).

## Service utilization

The percentage of clinic utilization by adult leprosy patients in the WP was 87.8%. Considering the utilization of the nearest clinic by patients, the majority had attended the nearest clinic to get leprosy treatment (n = 467, 69.5%) (Table 3). The most common reason for not attending the nearest clinic was to conceal the illness from known persons (n = 66, 32.2%). Of the patients, 32.2% (n = 66) believed that the faraway hospital provides better service and 24.8% (n = 51) wanted to continue treatment with the same specialist with whom the initial contact was made.

Ninety-two patients (13.7%) were hospitalized during their course of treatment. Of the 39 (42.5%) were admitted to the National Hospital, Colombo. The most common reason for hospital admission was type 2 reaction with fever and rash (n = 24, 26%) and to diagnose leprosy (n = 24, 26%). The majority were hospitalized for less than seven days (n = 62, 67.4%).

Considering the provision of field health services, half of the sample (n = 338, 50.3%) were knowledgeable of the location of the MOH office (Table 4). Of the sample, 401 patients (56.7%) indicated that the PHI visited their houses and gave health education (n = 363, 54%) by PHI. In about 552 patients (82.2%) all family contacts were screened for leprosy. Among them, family members of 299 (44.4%)patients were examined at the hospital, 207 (30.8%) by PHI and 46 (7%)were examined at MOH offices. Screening of the family members were not done among 120(17.8%)patients. Among the patients, 482 (71.7%) consented to a home visit

**Table 2. Distribution by delay of treatment initiation.**

| Levels of Delay | Mean(SD) | Median |
|---|---|---|
| Patient-related delay* | 16.8(32.9) | 8.0 |
| Time from presentation to a doctor to diagnosis** | 18.6(172.8) | 0.0 |
| Time from diagnosis to starting treatment** | 2.7(9.8) | 0.0 |
| Health care system delay** | 21.2(173.0) | 0.0 |
| Overall delay* | 17.5(33.2) | 8.0 |

*In months,

**In days

 

**Table 3. Distribution by utilization of treatment services (n = 672).**

| Description | Frequency | Percentage |
|---|---|---|
| **Attending to the nearest clinic (n = 672)** | | |
| Yes | 467 | 69.5 |
| No | 205 | 30.5 |
| **Reasons for not attending to the closest clinic (n = 205)** | | |
| To conceal the disease from known persons | 66 | 32.2 |
| Believing the current hospital provide a better service | 66 | 32.2 |
| To continue the treatment with the same specialist from the beginning | 51 | 24.8 |
| Lack of laboratory facilities | 14 | 6.8 |
| Other | 8 | 4.0 |
| **Hospitalized during the course of treatment** | | |
| Yes | 92 | 13.7 |
| No | 580 | 86.3 |
| **Reason for admission(N = 92)** | | |
| To diagnose leprosy | 24 | 26.0 |
| Fever with rash(Type 2 reaction) | 24 | 26.0 |
| Wounds | 09 | 9.8 |
| Dapsone induced hemolysis | 06 | 6.6 |
| Surgery | 06 | 6.6 |
| Numbness & swelling of the hands & feet | 05 | 5.5 |
| Other | 18 | 19.5 |
| **Period of hospitalization (N = 92)** | | |
| Less than seven days | 62 | 67.4 |
| 7–14 days | 19 | 20.6 |
| 15 days and above | 11 | 12.0 |

by health care workers. The main reason given by others for not permitting home visits by health care workers was to avoid neighbors knowing the disease condition (n = 167, 87.9%).

## Knowledge of leprosy

When assessing the knowledge of patients regarding the disease, 288 (42.9%) said, leprosy is more prone to develop in people who are living in overcrowded houses (Table 5). Leprosy can be transmitted by nasal droplets of an affected person was known by 526 (78.3%) patients. A majority (n = 524, 78%) knew that leprosy can be transmitted by closely living with an affected person who is not on treatment. More than half of the sample (n = 351, 52.2%) knew that, leprosy cannot be transmitted by living with a person who is on treatment. Out of the patients, 645 (96%) knew hypo pigmented anesthetic patches are associated with leprosy. Majority of the sample (n = 576, 85.7%) knew that leprosy can be completely cured by taking regular treatment. According to the method mentioned in the calculation of total knowledge score (which ranged between 5–100), more than half (57.9%, n = 389) of the study sample had a good and very good knowledge level. Mean knowledge score was 50.7 (SD = 17.9) (S1 Data).

## Discussion

The Sri Lankan government provides free health services for all citizens. Therefore, the majority of the Sri Lankan population rely on government health services [12,13]. This is reflected among leprosy patients in the present study; over half of the patients obtained initial

**Table 4. Distribution by the provision of field health services, patient perception and awareness on field services (n = 672).**

| Description | Frequency (n) | Percentage (%) |
|---|---|---|
| **MOH office has a role to provide services to leprosy-affected families** | | |
| Yes | 298 | 44.3 |
| No | 255 | 38.0 |
| *Not relevant | 119 | 17.7 |
| **Location of the MOH office** | | |
| Know | 338 | 50.3 |
| Don't know | 215 | 32.0 |
| *Not relevant | 119 | 17.7 |
| **Family members examined by** | | |
| Hospital | 299 | 44.4 |
| PHI | 207 | 30.8 |
| MOH office | 46 | 7.0 |
| Family member screening was not done | 120 | 17.8 |
| **PHI visited the house** | | |
| Yes | 401 | 56.7 |
| No | 245 | 36.5 |
| No, but contacted over the phone and gave advice | 26 | 3.8 |
| **Health education was given by PHI** | | |
| Yes | 363 | 54.0 |
| No | 309 | 46.0 |
| **Consented to home visit by health care workers** | | |
| Yes | 482 | 71.7 |
| No | 190 | 28.3 |
| **Reason for refusal of home visits (n = 190)** | | |
| To avoid neighbor's knowing the disease condition | 167 | 87.9 |
| Health workers may create unnecessary fear among family members | 15 | 7.9 |
| Others | 08 | 4.2 |

*Question is not relevant to patients from the Colombo Municipal Council (CMC) area

consultation at a government hospital. Once diagnosed, all patients have to attend government dermatology clinics since only government hospitals provide treatment for leprosy. Patients were referred to dermatology clinics from Out Patient Department (OPD)s of government and private hospitals, general practitioners, MOH's, and PHIs in the field and during School Medical Inspections (SMI). The number of patients referred by PHI may be comparatively low due to low contact examination coverage by PHIs, since the PHI is only able to examine males and their clinical knowledge and skills to identify disease may not be adequate. PHI is a grass root level public health officer who provides services in a vast health related areas apart from the disease notification and contact tracing. Being a male and not accompanying female with him during home visits, when he encounters a female leprosy contact, she is not examined and referred to a Medical Officer at MOH office or to a Dermatology clinic.

Patient-related delays occur not only due to patients' fault but also due to lack of experience by health care workers to detect the disease condition [14,15,16]. Health care staff coming across leprosy patient is a rare event since the total number of cases is low in Sri Lanka. Leprosy is confined to some areas of urban slums among low socio-economic community. Improving the awareness of leprosy of health staff and the community is important to

**Table 5. Distribution by knowledge of leprosy.**

| Description | Frequency (n = 672) | | |
|---|---|---|---|
| | **Correct** | **Incorrect** | **Don't know** |
| | **n (%)** | **n (%)** | **n (%)** |
| **Leprosy is more prone to develop in** | | | |
| People with malnutrition | 225(33.4) | 114(17.0) | 333(49.6) |
| People with poverty | 189(28.1) | 152(22.6) | 331(49.3) |
| People living in overcrowded houses | 288(42.9) | 89(13.2) | 295(43.9) |
| Poor personal hygiene | 248(36.9) | 94(14.0) | 330(49.1) |
| **Leprosy can be transmitted by** | | | |
| Nasal droplets of an affected person | 526(78.3) | 24(3.6) | 122(18.1) |
| Using the same toilet | 29(4.4) | 429(63.8) | 214(31.8) |
| Bath in the same well | 18(2.7) | 433(64.4) | 221(32.9) |
| Closely live with an affected person who is not on treatment | 524(78.0) | 26(3.9) | 122(18.1) |
| Genetically | 47(7.0) | 189(28.1) | 436(64.9) |
| Living with a person who is on treatment | 62(9.3) | 351(52.2) | 259(38.5) |
| **Leprosy patient can present with** | | | |
| Hypo pigmented anesthetic patches | 645(96.0) | 4(0.6) | 23(3.4) |
| Nodules over the skin | 137(20.4) | 17(2.5) | 518(77.1) |
| Cough | 43(6.4) | 55(8.2) | 574(85.4) |
| Muscle weakness | 76(11.3) | 23(3.4) | 573(85.3) |
| Disability | 107(15.9) | 19(2.8) | 546(81.3) |
| **Other** | | | |
| Ability to involved with social activities while on treatment | 619(92.1) | 23(3.4) | 30(4.5) |
| Leprosy leads to vision impairment in some patients | 320(47.6) | 72(10.7) | 280(41.7) |
| Leprosy does not affect the nerve function | 129(19.2) | 276(41.1) | 267(39.7) |
| Leprosy leads to disability and disfigurement if left untreated | 554(82.4) | 41(6.1) | 77(11.5) |
| Leprosy can be completely cured by taking regular treatment | 576(85.7) | 33(4.9) | 63(9.4) |

overcome this problem [17,18]. Usually, at government hospitals, MDT drugs are started on the same day, for patients who can be diagnosed clinically followed by Slit Skin Smear test. If the diagnosis is doubtful, the clinic staff has to take a biopsy and wait for the results. The time duration for this will vary between 2–6 weeks in different hospitals. Health care system delay arises due to this process. If the first biopsy report is inconclusive, a further period is required for the second biopsy.

Some of the Dermatology clinics in the government hospitals are overcrowded with patients. Furthermore, dermatology clinics provide services to leprosy as well as to other dermatological conditions. Usually, the hypo pigmented patches arise due to leprosy are pain free/symptomless. Therefore, patients are reluctant to spend a half day in a dermatology clinic to diagnose it, which leads to patient related delay.

A study carried out by Nicholos et al [19] in India (Purulia) and Bangladesh (Nilphamari), in patients currently receiving treatment for leprosy revealed; delay estimated from time of the first symptoms to start of effective treatment had a mean of 18 months (median nine months) in Purulia and mean of 20 months (median 12 months) in Nilphamari. In the current study, the mean delay was 17.5 months (median of 8 months) which shows similar results. This reflects the poor community awareness of leprosy symptoms in South Asian countries. This is significant since many patients with delays in the presentation will end up with deformities. Furthermore, Zhang et al. [8] carried out a study in China, and the total mean delay was 50·2

months (median 36 months). The mean patient delay was 24·4 months (median 9·5 months) and the mean health service delay was 25·7 months (median 12 months). In the current study mean total delay was 17.5 months (median eight months), mean patient-related delay 16.8 months (median of eight months) and mean health care system delay was 21.2 days, which are less than the Chinese study. In a study carried out by Deps et al. in Brazil [20] the mean delay in MB (27·2 months) was greater than in PB (21·3 months). The current study shows a similar pattern but with lower duration (mean delay of the MB patients 18.5 months and PB patients it was 15.2 months). These findings reflect an increase in awareness and health-seeking behavior of the Sri Lankan population compared to the Chinese and Brazilian population.

However, according to the Sri Lanka Anti Leprosy Campaign data 2015, 44.7% of patients presented to the clinic as late presentations (>6 months). Percentage of early diagnosis (< 6 months) of cases in Colombo, Gampaha and Kalutara district in 2015 were 43%, 48% and 48% respectively [9]. These statistics means nearly half of the patients present late to the clinics in Sri Lanka.

Clinic utilization by Paucibacillary patients was 87.9% (n = 199) and in Multibacillary patients it was 87.7% (n = 473). Clinic utilization of both categories was more or less similar and satisfactory. A study carried out in India by Renita et al. [21] on health care utilization by leprosy patients revealed the clinic utilization was 58·1%(n = 115). In the current study, clinic utilization was higher than the Indian study. It reflects the increased health-seeking behavior of the Sri Lankan population. This concept highlights that once diagnosed, educated and counseled properly, the patients tend to attend a clinic and continue the treatment except those with poor family support and special cases such as drug addicts, prisoners etc.

As stated above, the Sri Lankan health care system is well established and people trust the government healthcare service. Therefore, a majority of patients (69.5% (n = 467)) utilized the nearest government clinic. Furthermore, basic diagnostic facilities are available in the majority of the Dermatology clinics. Nevertheless, only half of the sample knew the location of the MOH office. These findings indicate that only half of the patients were aware of field health services. Nearly one-third of patients (n = 190, 28.3%) were refusing of home visits by health care workers.

Some of the patients were hiding their disease from family members due to stigma. Moreover, they refused visit by the health workers due to fear of revealing the disease. In such instances, range PHI of the area organized a skin clinic close to patient's residence to cover the community in the area, including patient family members. However this highlights the importance of a need for a mechanism developed to screen family contacts.

The study had some limitations. The study participants had been on continuous treatment for some time. As a result of that, they were continuously getting health education from the hospital staff regarding the disease. This may be one reason of them having a high knowledge level. Furthermore, the study findings are applicable only to the Western province of Sri Lanka. It cannot be generalized to the other parts of the country.

## Conclusion

Clinic utilization by adult leprosy patients in the WP was satisfactory. There was a long patient-related delay in attending a hospital and a considerable health system delay for starting treatment. More than half of the study sample had a good knowledge of disease transmission, symptoms and complications of the disease. Nearly half of the sample was aware of the services provided to leprosy patients at the MOH office and by PHI. Although contact screening was satisfactory, nearly one third of patients did not consent to a home visit by health care workers. Majority of the contacts were examined at hospital clinics.

For early disease identification, it is recommended to conduct regular awareness and training to GPs, OPD Doctors, MOH's and field health workers (such as PHIs and PHMs). To minimize patient-related delay, a community awareness program including mass media campaign should be conducted. Mandatory contact tracing by strengthening the legislative aspect and reinforce contact screening activities in all MOH offices will improve the number of cases diagnosed further.

## Supporting information

**S1 Text. Study Questionair on Utilization of healthcare services by leprosy patients.**
(DOCX)

**S1 Data. Data set.**
(XLSX)

## Acknowledgments

The authors would like to express gratitude to the experts who gave support to plan this study and health staff in the dermatology clinics and MOH offices and, data collectors.

## Author Contributions

**Conceptualization:** Nadeeja Roshini Liyanage, Mahendra Arnold, Supun Wijesinghe.

**Data curation:** Nadeeja Roshini Liyanage, Supun Wijesinghe.

**Formal analysis:** Nadeeja Roshini Liyanage, Mahendra Arnold, Supun Wijesinghe.

**Investigation:** Supun Wijesinghe.

**Methodology:** Nadeeja Roshini Liyanage, Mahendra Arnold, Supun Wijesinghe.

**Project administration:** Nadeeja Roshini Liyanage.

**Resources:** Nadeeja Roshini Liyanage.

**Software:** Nadeeja Roshini Liyanage.

**Supervision:** Mahendra Arnold.

**Validation:** Mahendra Arnold, Supun Wijesinghe.

**Visualization:** Mahendra Arnold.

**Writing – original draft:** Nadeeja Roshini Liyanage.

**Writing – review & editing:** Mahendra Arnold, Supun Wijesinghe.

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
