## [Decision Letter · Decision Letter 0]

2 Oct 2020

Dear Dr. Liyanage,

Thank you very much for submitting your manuscript "Utilisation of Government Healthcare Services by Adult Leprosy Patients in the Western 

Province, Sri Lanka" for consideration at PLOS Neglected Tropical Diseases. As with all papers reviewed by the journal, your manuscript was reviewed by members of the editorial board and by several independent reviewers. The reviewers appreciated the attention to an important topic. Based on the reviews, we are likely to accept this manuscript for publication, providing that you modify the manuscript according to the review recommendations. 

Sincerely,

Linda B Adams

Associate Editor

Gerson Penna

Deputy Editor

Reviewer's Responses to Questions

**Key Review Criteria Required for Acceptance?**

**Methods**

-Are the objectives of the study clearly articulated with a clear testable hypothesis stated?

-Is the study design appropriate to address the stated objectives?

-Is the population clearly described and appropriate for the hypothesis being tested?

-Is the sample size sufficient to ensure adequate power to address the hypothesis being tested?

-Were correct statistical analysis used to support conclusions?

-Are there concerns about ethical or regulatory requirements being met?

Reviewer #1: This study covers an important area in diagnosis of leprosy, namely the delay in diagnosis of leprosy. since delay in diagnosis can directly affect subsequent disability in leprosy the study is of importance and relevance to NTD s.

Reviewer #2: The authors clearly state their primary objective to determine the utilization of leprosy health services by patients. However, they do not discuss their secondary objective of testing patient's leprosy knowledge until the methods section. The authors should discuss both objectives in the abstract and introduction sections and especially discuss the role that general knowledge or lack of knowledge about leprosy may play in impacting the accessibility of health services. The authors describe the study population adequately although it is slightly unclear about the role of the clinic's leprosy register in patient recruitment. Otherwise, the study design, sample size, and analyses were well done and properly addressed the objectives.

Reviewer #3: Yes

**Results**

-Does the analysis presented match the analysis plan?

-Are the results clearly and completely presented?

-Are the figures (Tables, Images) of sufficient quality for clarity?

Reviewer #1: results are in keeping with he analysis plan. The results are clearly mentioned. I have requested one clarification of a statement.

Reviewer #2: Analyses and the results were explained thoroughly. All of the tables were well designed and easy to understand and interpret. Only correction in the Results was that at several points the authors refer to a "Type 11" reaction, which is confusing. I would recommend correcting that to be "Type 2" to improve clarity.

Reviewer #3: Yes

**Conclusions**

-Are the conclusions supported by the data presented?

-Are the limitations of analysis clearly described?

-Do the authors discuss how these data can be helpful to advance our understanding of the topic under study?

-Is public health relevance addressed?

Reviewer #1: the conclusions are clearly mentioned and supported by the data presented.

Reviewer #2: The conclusions presented are supported by the data presented, however the authors need to further expand on the significance of these conclusions both for Sri Lanka and for the global leprosy field. Additionally, while the authors offer informative comparisons of their results to the results of studies conducted in other countries, they do not discuss any limitations of their own study.

Reviewer #3: Yes

**Editorial and Data Presentation Modifications?**

Reviewer #1: the Author summary was not clearly presented. I have taken the liberty to make some modifications. The modified version is attached.

Reviewer #2: There are a few minor grammatical errors throughout the manuscript. Additionally, the second half of the author summary is a little awkward. I understand what the authors are trying to say, but I think a few sentences could be reworded to improve clarity.

Reviewer #3: (No Response)

**Summary and General Comments**

Reviewer #1: The study is of relevance since delay in diagnosis of leprosy is of importance to all endemic countries.

Reviewer #2: Overall, this is a well-written manuscript of an interesting cross-sectional study of healthcare utilization by leprosy patients in the Western Province of Sri Lanka. The results are interesting, and the study was thoroughly conducted. However, I felt that the authors needed to expand on their discussion of how these results affect both Sri Lanka and the field of leprosy research. The authors started to do this with an interesting comparison of their results to other similar studies conducted in other countries. However, they did not discuss why their results were better or worse than those seen elsewhere. For instance, what programs or healthcare factors affect these outcomes? More discussion on these factors could really help to show why their results are relevant both in Sri Lanka and also globally.

Reviewer #3: (No Response)

PLOS authors have the option to publish the peer review history of their article (what does this mean?). If published, this will include your full peer review and any attached files.

Reviewer #1: No

Reviewer #2: No

Reviewer #3: Yes: Srinivas G
---

## [Editor Report · Decision Letter 1]

10 Nov 2020

Dear Dr. Liyanage,

We are pleased to inform you that your manuscript 'Utilization of Government Healthcare Services by Adult Leprosy Patients in the Western Province, Sri Lanka' has been provisionally accepted for publication in PLOS Neglected Tropical Diseases.

Best regards,

Linda B Adams

Associate Editor

Gerson Penna

Deputy Editor

---

## [Editor Report · Acceptance letter]

21 Dec 2020

Dear Dr. Liyanage,

We are delighted to inform you that your manuscript, "Utilization of Government Healthcare Services by Adult Leprosy Patients in the Western 
Province, Sri Lanka," has been formally accepted for publication in PLOS Neglected Tropical Diseases.

Best regards,

Shaden Kamhawi

co-Editor-in-Chief

Paul Brindley

co-Editor-in-Chief
